

# Negligible Temperature Dependence of the Ozone-Iodide Reaction and Implications for Oceanic Emissions of Iodine

Lucy V. Brown[1], Ryan J. Pound[1], Lyndsay S. Ives[1], Matthew R. Jones[1], Stephen J. Andrews[1], and Lucy J. Carpenter[1]

[1]Wolfson Atmospheric Chemistry Laboratories, Department of Chemistry, University of York, York, YO10 5DD, UK.

**Correspondence:** Lucy V. Brown (lvb520@york.ac.uk)

**Abstract.** The reaction between ozone and iodide is one of the main drivers of tropospheric ozone deposition to the ocean, due to the ubiquitous presence of iodide in the ocean surface and its rapid reaction with ozone. Despite the importance of this sea surface reaction for tropospheric ozone deposition, and also as the major source of atmospheric iodine, there is uncertainty in its rate and dependence on aqueous phase temperature. In this work, the kinetics of the heterogeneous second order reaction between ozone and iodide were investigated using conditions applicable to coupled ocean-atmosphere systems ($1 \times 10^{-7}$ – $1 \times 10^{-5}$ M [iodide], 40 ppb ozone, 288 – 303 K, 15.0 psi). The Arrhenius parameters determined of A = $1.0 \pm 4.6 \times 10^{11}$ $M^{-1} s^{-1}$ and Ea = $8.5 \pm 10.9$ $kJ\,mol^{-1}$ show that the reaction has a negligible positive temperature dependence, which could be weakly negative within errors. This is in contrast to a previous study that found a strong positive activation energy and a pre-exponential factor many orders of magnitude greater than determined here. The re-measured kinetics of ozone and iodide were used to constrain a state-of-the-art sea surface microlayer (SML) model. The model replicated results from a previous laboratory study of the temperature dependence of hypoiodous acid (HOI) and molecular iodine ($I_2$) emissions from an ozone-oxidised iodide solution. This work has significance for global modelling of dry deposition of ozone to the ocean and the subsequent emissions of iodine-containing species, thus improving understanding of the feedbacks between natural halogens, air quality, and climate change.

## 1 Introduction

Ozone ($O_3$) is an important tropospheric pollutant and greenhouse gas, with estimated radiative forcing of 0.40 $Wm^{-2}$ (reported ranges span 0.20 - 0.65 $Wm^{-2}$) (R.T. Watson et al., 1995; Ramaswamy et al., 2001; Forster et al., 2007; Myhre et al., 2013). Ozone concentrations must be monitored and accurately predicted, as short term exposure can harm the respiratory and cardiovascular systems in humans, while long term exposure is thought to contribute to respiratory mortality and new-onset asthma in children (Nuvolone, 2018). Tropospheric ozone also causes risk to global food production due to its phytotoxic effects on crop plants, including destruction of photosynthetic pigments and decrease in crop growth and productivity (Rai and Agrawal, 2012). The tropospheric ozone burden ($\sim$340 Tg) is controlled by a balance of influx from the stratosphere ($\sim$550 Tg ($O_3$) year$^{-1}$), chemical production ($\sim$5100 Tg ($O_3$) year$^{-1}$), chemical destruction ($\sim$4650 Tg ($O_3$) year$^{-1}$), and dry deposition ($\sim$1000 Tg ($O_3$) year$^{-1}$) to the Earth's surface (Stevenson et al., 2006).





Oceanic dry deposition, though considerably slower than deposition to crops and soil (Wesely and Hicks, 2000), is influential in global atmospheric models due to the significant coverage of the Earth by oceans. Dry deposition of ozone to the ocean surface is estimated to contribute approximately a quarter of the total loss of ozone from the atmosphere and represents the single largest deposition flux of ozone (mean of 361 Tg ($O_3$) year$^{-1}$, calculated from 15 different models) (Hardacre et al., 2015).

The dry deposition velocity ($v_d$) of ozone to the ocean has been reported by several authors with highly variable results, but typically of the order of 0.01 - 0.10 cm s$^{-1}$ (Galbally and Roy, 1980; Garland et al., 1980; Wesely et al., 1981; Lenschow et al., 1982; Kawa and Pearson Jr., 1989; McKay et al., 1992; Heikies et al., 1996; Gallagher et al., 2001; Whitehead et al., 2009; McVeigh et al., 2010; Helmig et al., 2012). In lieu of accurate parameterisation methods, global ozone models typically apply a single global deposition velocity of 0.05 cm s$^{-1}$, despite large variances in measured values (Ganzeveld et al., 2009).

To understand deposition of gas to solution, the process is often thought of analogously to electrical resistance. While gaseous deposition to solution is governed by a series of complex processes, the transfer is often simplified by separating the main processes into individual resistances. The total resistance ($r_t$) to deposition is the inverse of the deposition velocity. It is comprised of three terms (Eq. 1), where $r_a$ is the aerodynamic resistance due to atmospheric turbulence, influenced by factors such as wind speed and aerodynamic roughness (Chang et al., 2004), $r_b$ is the gas phase film resistance, or diffusion across

the quasilaminar sub-layer directly above the surface, and $r_s$ is the surface resistance. The latter is of the greatest significance when considering chemical controls on deposition of ozone, accounting for >90 % of the total observed resistance over marine waters (Lenschow et al., 1982; Kawa and Pearson Jr., 1989). In this work, $r_a$ and $r_b$ (Eq. 1) were combined into one air resistance term, termed $r_a$ herein, allowing isolation of $r_s$. Under conditions where turbulence is negligible, i.e. diffusion processes dominate, $r_s$ can be predicted by Eq. 2, (Garland et al., 1980), showing that it is dependent on the second order rate constant

for the reaction between ozone and any ozone reactive species, i ($k_i$), as well as the dimensionless Henry's law coefficient, $H$, the concentration of species i, [i], and the aqueous diffusivity of ozone, $D_{aq}$.

$$v_d = \frac{1}{r_t} = \frac{1}{r_a + r_b + r_s} \qquad (1)$$

$$r_s = \frac{H}{\sqrt{D_{aq} \sum k_i[i]}} \qquad (2)$$

For oceanic dry deposition of ozone, gas transfer is fast but the solubility of ozone is low, therefore deposition is thought to be largely driven by chemical reactions. One of the most significant reactions governing oceanic ozone deposition is its reaction with iodide (R1). Several studies have demonstrated that oceanic concentrations of iodide enhance the deposition velocity of ozone (Oh et al., 2008; Ganzeveld et al., 2009; Coleman et al., 2010; Sarwar et al., 2016). Sarwar et al. (2016) quantified the effect of iodide in seawater as a 0.023 cm s$^{-1}$ enhancement in ozone deposition velocity ($v_d$), or an increase in median

modelled oceanic $v_d$ over the Northern Hemisphere from 0.007 cm s$^{-1}$ with no explicit chemical effect applied, to 0.030 cm s$^{-1}$ when ozone-iodide interactions are included. Iodide concentrations at the sea surface are positively correlated with temperature (Chance et al., 2014; MacDonald et al., 2014; Sherwen et al., 2019). It should however be noted that at higher



temperatures, deposition may become limited by reduced ozone solubility in water, thereby minimising the impact of iodide in tropical and sub-tropical regions (Ganzeveld et al., 2009).

$$O_3 + I^- \longrightarrow [OOOI]^- \longrightarrow IO^- + O_2 \qquad\qquad (R1)$$

The mechanism of the ozone-iodide reaction depends upon the surrounding conditions, specifically the concentrations of iodide and ozone, thus experimental studies can potentially differ in their conclusions depending on their choice of these parameters (Moreno et al., 2018). At iodide concentrations below $\sim 10^{-5}$ M, as found in oceanic systems (Chance et al., 2019), the reaction with ozone is thought to occur in the bulk aqueous phase, where ozone is dissolved into solution before reacting.
The alternative is surface reactivity following Langmuir-Hinshelwood kinetics, which occurs at higher concentrations of iodide (Moreno et al., 2018; Moreno and Baeza-Romero, 2019). Further, it is known that many other ocean-relevant species, particularly organic compounds, react with ozone via a Langmuir-Hinshelwood mechanism, e.g. chlorophyll-a, polyunsaturated fatty acids such as linoleic acid and oleic acid, and polycyclic aromatic hydrocarbons (PAHs) such as naphthalene, anthracene and pyrene (Mmereki and Donaldson, 2003; Mmereki et al., 2004; Donaldson et al., 2005; Raja and Valsaraj, 2005; Clifford et al.,
2008; Zhou et al., 2014).

To quantify the impact of the ozone-iodide reaction on oceanic dry deposition of ozone, the concentration of iodide [I$^-$] and second order rate constant for the ozone-iodide reaction, $k_{I^- - O_3}$, must be known. Iodide concentrations in the surface ocean have been measured by several authors, and are typically within the range of 20-200 nM (De Souza and Sen Gupta, 1984; Campos et al., 1996, 1999; Chance et al., 2014), with Chance et al. (2014) reporting a median of 74 nM and interquartile range
of 27 to 135 nM at the sea surface.

The second order rate constant for the reaction between ozone and iodide, $k_{I^- - O_3}$, has been measured in the past, under a range of conditions (Garland et al., 1980; Hu et al., 1995; Magi et al., 1997; Liu et al., 2001; Rouvière et al., 2010; Shaw and Carpenter, 2013). Published $k_{I^- - O_3}$ values, measured at room temperature, range between $1 - 4 \times 10^9$ M$^{-1}$ s$^{-1}$. Only one previous study has investigated the temperature dependence of this reaction, obtaining a strong positive dependence with
temperature (A = $1.4 \times 10^{22}$ M$^{-1}$ s$^{-1}$ and Ea = 73.08 kJ mol$^{-1}$, with an estimated error of 40% (Magi et al., 1997)), although this study was carried out under conditions which could promote surface reactivity. It is important to determine the temperature dependence of this reaction under conditions which favour the bulk reaction. Accurate measurement of the temperature dependence of the ozone-iodide reaction will allow better understanding and prediction of global ozone deposition and iodine-containing emissions.

The reaction between ozone and iodide leads to emissions of hypoiodous acid (HOI) and I$_2$ according to reactions R2 - R6. Emissions of gaseous iodine have significant impacts on tropospheric ozone. Photolysis of gaseous iodine species produces atomic iodine (I) which is rapidly oxidised to IO by ozone. IO is lost by reaction with HO$_2$ to re-form HOI. Photocycling of iodine-containing species therefore leads to efficient destruction of ozone in the troposphere (Read et al., 2008; Saiz-Lopez and Von Glasow, 2012). Additionally, recent measurements have revealed the presence of iodine in the lower stratosphere,
contributing to stratospheric ozone loss, primarily via heterogeneous chemistry occurring on particles (Koenig et al., 2020).



Understanding the temperature dependence of the reaction between ozone and iodide is therefore also important for understanding ozone loss in the low temperatures of the stratosphere.

$$IO^- + H^+ \rightleftharpoons HOI \tag{R2}$$

$$HOI + I^- \rightarrow HOI_2^- \tag{R3}$$

$$HOI_2^- + H^+ \rightarrow I_2 + H_2O \tag{R4}$$

$$HOI_{(aq)} \rightleftharpoons HOI_{(g)} \tag{R5}$$


$$I_{2(aq)} \rightleftharpoons I_{2(g)} \tag{R6}$$

It is clear that knowledge of the kinetics of the reactions of ozone with iodide is essential for understanding reactivity at the sea surface and, in particular, required for accurate modelling of ozone dry deposition to the ocean and subsequent emissions of iodine-containing species. In this work, the second order rate constant of the reaction between ozone with iodide and its associated temperature dependence were measured. Our study employed conditions which emulated oceanic reactivity of iodide, i.e. low concentrations of iodide and ozone, to target bulk reactivity. We then use this kinetic knowledge to explore previous lab studies of iodine-containing emissions using a recently developed coupled-chemistry ocean atmosphere exchange model.

## 2 Methods

### 2.1 Experimental Method

#### 2.1.1 Sample Preparation

Ozone was generated by a Pen-Ray ultra-stable ozone generator (97-0067-02, UVP), and the concentration adjusted by moving the lamp jacket. The flow was then diluted by dried, hydrocarbon scrubbed compressed air (lab generated). The ozone concentrations introduced into the flow reactor were measured by bypassing the flow reactor, with detection by a commercial UV photometric ozone analyser (model 49i, Thermo). The primary experimental media was 10 mM phosphoric acid ($H_2PO_4$ ; Sigma-Aldridge, 98.5 – 101%) in pure water (HPLC grade, Fisher Chemical) at pH 8, attained through small volume additions of 20% NaOH. The primary media was then ozonized to remove ozone-reactive contaminants. After ozonisation and





blank measurement, the primary solutions were spiked with an iodide standard (5 mM) to the desired concentration. Potassium iodide (KI; 99% purity, Fluorochem) standards were gravimetrically prepared in, and subsequently diluted by, ultra-pure
deionised water (18.2 mΩ).

### 2.1.2   Gas Flow Control

A temperature-controlled kinetic heterogeneous flow reactor (Fig. 1) was designed in order to enable variable flow rates of ozone, and hence variable exposure times, over iodide solutions. Hydrocarbon-filtered dry air was separated into three flows; ozonized air controlled by mass flow controller 1 (MFC 1, Alicat, MC-10SLPM-D/CM, CIN), a diluent flow (MFC 2, Alicat, MCP-50SLPM-D/5M) and a third flow diverted to a Nafion dryer (MFC 3, Aalborg, GFC17). The combination of flows from
MFC1 and MFC2 enabled the generation of a large total flow of ozone-enriched air with a constant ozone concentration. This ozone enriched air was passed either through the flow reactor or a bypass line (MFC 4, Alicat, pc-30PSIA-D-PCV65/5P); any excess was removed through a back pressure regulator (BPR 1, Alicat, PCP-100PSIG-D/5P). Downstream of the flow tube the gas was dehumidified using a Nafion dryer (Perma Pure, MD-110-12F-4) and analysed by the ozone monitor ($\sim 1.4$ SLPM).
Gas surplus to the analytical requirement was removed from the system by BPR 2 (15.0 psi), which was a modified MFC (Alicat, MCP-10SLPM-D/CM). This was modified by using absolute pressure as the process variable, and moving the valve downstream of the pressure sensor. The valve action was then inverted, meaning an increase of pressure in the flow reactor would cause an opening of the valve, allowing constant pressure to be maintained. All ozone-containing gas was passed through a charcoal scrubber prior to venting. The ozone monitor was logged using DAQFactory, and all Alicat MFCs and BPRs were controlled and logged with DAQFactory (Azeotech).

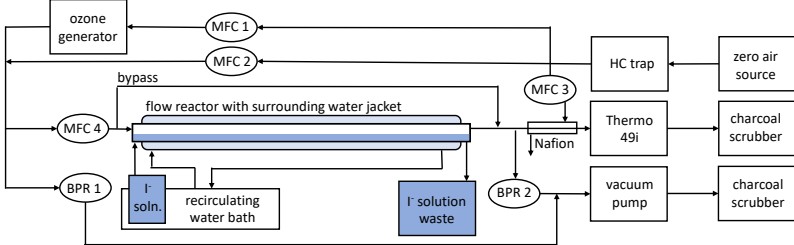

**Figure 1.** Schematic of the experimental set up. MFC = mass flow controller, BPR = back pressure regulator.


### 2.1.3   Temperature and Fluid Control

The flow reactor was temperature controlled by a water jacket, supplied by a recirculating water bath chiller (TX150 and R3, Grant Instruments). The iodide or blank solutions were held within the reservoir of this water bath to equilibrate their temperature. In order to minimise any depletion of iodide from the solution during exposure to ozone, it was continuously





pumped into the flow reactor using a peristaltic pump (100 series, Watson Marlow) via chemically resistant flexible tubing (Marprene, Watson Marlow). Once passed through the flow tube, the iodide solution drained into a sealed, pressure equilibrated waste bottle.

To estimate the iodide depletion during the course of the experiment with no replenishment, the rate of loss of iodide, $d[I-]/dt$, was calculated from the second order rate constant for ozone and iodide and the molar concentrations of iodide and ozone in

the reacto-diffusive layer of ozone (Eq. 3). In these calculations $k_{I^--O_3}$= $1.2 \times 10^9$ M$^{-1}$s$^{-1}$ was used (Liu et al., 2001). The reacto-diffusive depth of ozone, $\delta$, is the thickness of the layer in which the ozone-iodide reaction can occur (Davidovits et al., 2006), calculated by Eq. 4. $D_{aq}$ is the molecular diffusivity of ozone in water ($1.90 \times 10$-9 m$^2$ s$^{-1}$ at 298 K), calculated using the temperature-dependent relationship Eq. 5, where $T$ is the temperature in Kelvin (Johnson and Davis, 1996). The reacto-diffusive depth multiplied by the surface area of the liquid this gives the liquid volume in the flow reactor in which

ozone is available for reaction, $V_\delta$ (Eq. 6). Assuming a worst-case scenario where mechanical turbulence from stirrer bars was not sufficient to replenish any iodide from the bulk solution into the reacto-diffusive layer, the total molar quantity of iodide available for reaction (I$_\delta^-$) was calculated by Eq. 7. Total loss of iodide during the total experiment time was calculated by Eq. 8 and the potential percent loss of iodide was therefore calculated by Eq. 9. Where the expected loss of iodide was greater than 10% the solution was pumped through the flow reactor sufficiently fast to give a residence time (Eq. 10) which, when applied

in Eqs. 8 and 9 gave a I$^-$ percentage loss of < 10%. Pump rates therefore varied with iodide concentration, however it was verified experimentally that pump rate did not affect deposition velocity within the flow tube.

$$\frac{d[I^-]}{dt} = -k_{I^--O_3}[O_3][I^-] \tag{3}$$

$$\delta = \sqrt{\frac{D_{aq}}{k_{I^--O_3}[I^-]}} \tag{4}$$


$$D_{aq} = 1.10 \times 10^{-6} exp(\frac{-1896}{T}) \tag{5}$$

$$V_\delta = \delta \times SA \tag{6}$$

$$I_\delta^- = [I^-] \times V_\delta \tag{7}$$

$$I^- \, loss \, (absolute) = \frac{d[I^-]}{dt} \times total \, experiment \, time \tag{8}$$



$$\% \, loss \, I^- = \frac{I^- \, loss \, (absolute)}{I_\delta^-} \times 100 \qquad (9)$$


$$residence \, time = \frac{liquid \, volume}{pump \, rate} \qquad (10)$$

### 2.1.4 Measurement of Aqueous Iodide

Iodide concentrations in the reservoir and the waste stream were directly quantified using UV-vis spectrophotometry at 226 nM following ion exchange chromatography (IC) (Jones et al., 2023). The IC used a Dionex IonPac AS23 guard and analytical

column (4 x 250 mm), with a mobile phase eluent of 0.4 M NaCl flowing at $0.64 \, \mathrm{ml \, min^{-1}}$. The sample injection volume was 400 µl, run time 16.1 minutes, and iodide was detected at 11.8 minutes. Samples were frozen at the time of experiment and defrosted prior to batch analysis.

### 2.2 Determining Surface Resistance and Ozone Uptake

To measure surface resistance, ozone-containing gas ($[O3]_{starting}$ = 40 ppb) was passed over the buffered blank solution or

iodide solutions, that were pumped through the reactor at liquid flow rates of between $9 - 35 \, \mathrm{ml \, min^{-1}}$. Gaseous flow rates were set at 1600, 1900, 2400, 3200 and 4500 SCCM, giving ozone-solution reaction times ranging from $24 - 66$ s (reaction time = flow rate/headspace volume). Blanks were performed in duplicate or triplicate, and measurements over iodide in triplicate. An example experiment output is shown in Fig. 2.

Residual ozone was measured after each reaction time; and a mean ozone concentration for each reaction time was obtained

([O3]). A plot of ln[O3] against reaction time (Fig. 3) yielded a linear trend, the gradient of which was calculated for each repeat of both the blank ($m_{blank}$), and the iodide-containing samples ($m_{sample}$), which were each averaged. A blank corrected gradient ($m_{corrected}$, Eq. 11) was used to calculate $v_d$ by Eq. 12, where $V$ is the headspace volume and $SA$ is the liquid surface area (values for all physical constants and meta data provided in Appendix A and calculations described in Appendix B). Deposition velocity was corrected for gas diffusion limitations at this stage, as described in Appendix C.

Total resistance, $r_t$, is the inverse of gas-phase corrected $v_d$, from which $r_s$ can be calculated (Eq. 1). $r_a$ is variable in the environment, but constant in the controlled environment of the flow reactor. Measured over a high iodide concentration (0.02 M), it is assumed that there is negligible surface resistance, therefore $1/r_t \approx 1/r_a$ (Galbally and Roy, 1980). When corrected for gas diffusion limitations (Appendix C), an $r_a$ of $0.48 \pm 0.02 \, \mathrm{s \, cm^{-1}}$ was obtained. This value was subtracted from total resistance measured for each sample.

$$m_{corrected} = m_{sample} - m_{blank} \qquad (11)$$

$$v_d = \frac{-m_{corrected}V}{SA} \qquad (12)$$



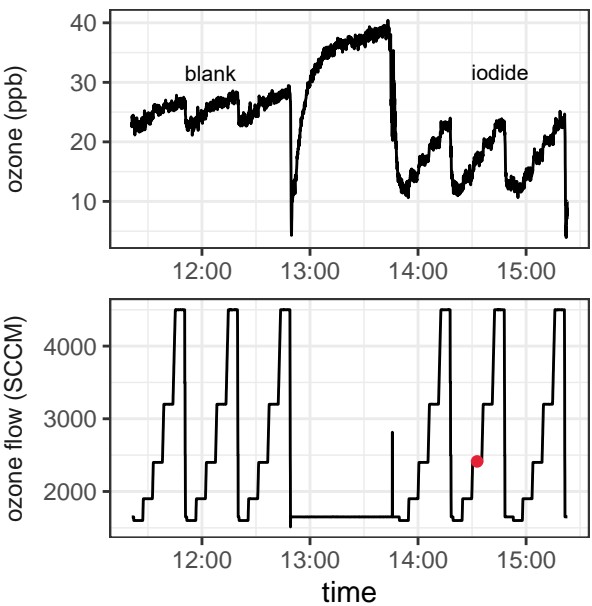

**Figure 2.** Experimental output for a typical measurement, including residual ozone measured downstream of the flow tube and the concurrent flow rate of ozone over the solution. A blank measurement and the measurement following iodide spiking are shown. The red filled circle (●) indicates the timing of the collection of the iodide "midpoint" sample from the waste stream. Experimental conditions: $T$ = 303 K, $[I^-]$ = 633 nM, phosphate buffer (10 mM, pH = 8), $[O_3]_{starting}$ = 40 ppb.

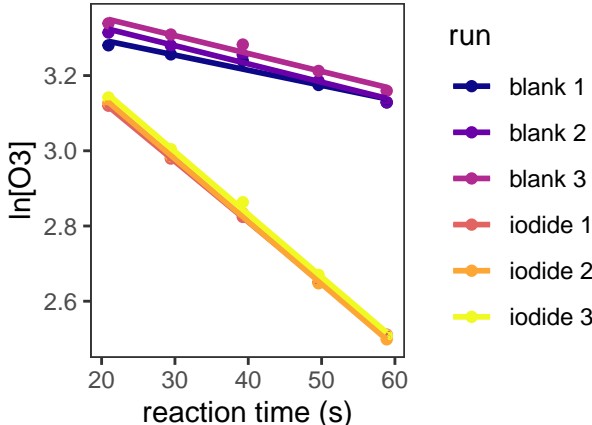

**Figure 3.** ln[O3] against reaction time for experimental conditions: $T$ = 303 K, $[I^-]$ = 633 nM, phosphate buffer (10 mM, pH = 8), $[O_3]_{starting}$ = 40 ppb.



The relationship between the second order rate constant, $k_{I^--O_3}$, and $r_s$ was defined by Eq. 2, where $H$ is the temperature-dependent dimensionless Henry's law constant of ozone (3.63 at 298 K), calculated using Eq. 13 (Kosak-Channing and Helz,

1983), $T$ is the solution temperature (K) and $\mu$ is the molar ionic strength ($\approx$ molar concentration, at the concentrations used in this work). To apply Eq. 2 to our measurements, for each temperature applied, a plot of $1/r_s$ against $\sqrt{[\text{I-}]}$ gave a linear relationship (see Results and Discussion, Fig. 5), from which the gradient, $m$, was used to calculate $k_{I^--O_3}$, according to Eq. 14.

$$lnK_h = -2297T^{-1} + 2.659\mu - 688.0\mu T^{-1} + 12.19 \tag{13}$$


$$k_{I^--O_3} = \frac{(mH)^2}{D_{aq}} \tag{14}$$

## 3    Results and Discussion

### 3.1    Kinetics and temperature dependence of the ozone-iodide reaction

Conditions within the flow reactor were chosen to emulate the remote marine surface ocean and atmospheric boundary layer.

The mixing ratio of ozone in air was not expected to affect ozone uptake to iodide solution, as under these conditions a bulk aqueous reaction between ozone and iodide is anticipated, for which a lack of ozone dependence is characteristic. If the reaction were occurring as a surface-mediated Langmuir-Hinshelwood reaction, an exponential increase of uptake at lowering ozone concentrations would be observed. Under the conditions of our experiments, there was no significant dependence of deposition velocity on ozone mixing ratio (Fig. 4, p = 0.74), confirming the reaction occurs in the bulk phase under the conditions

employed. Despite the lack of dependence of ozone mixing ratio on reactivity, consistent conditions of 40 ppb ozone were used in each experiment to mimic a typical mixing ratio of ozone in the troposphere. Similarly, although there is evidence that pH has no impact on ozone deposition to iodide solutions (Schneider et al., 2022), the solutions were buffered to pH 8 to mimic typical oceanic alkalinity.

Iodide was the reagent in excess in the pseudo-first order conditions sought for kinetic analysis. At the low iodide concen-

trations required to emulate marine conditions, iodide was expected to be depleted during the $\sim$90-minute experiment time (Schneider et al., 2020), therefore the liquid phase in the flow reactor was continuously replenished from a reservoir, passing through the flow reactor and out to waste. To verify that the chosen pump rate was sufficient to keep iodide depletion below 10% during the residence time of the liquid in the reactor, liquid samples were collected before and after being exposed to ozone in the flow reactor. The sample after the flow reactor was taken at the mid-point of the experiment (ozone flow rate

3 of 5, during run 2 of 3, indicated by the red diamond on Fig. 2). For all reported experiments, the iodide concentration after the experiment was verified by liquid chromatography. It was confirmed that the concentration during the mid-point of the experiment, taken to represent average iodide loss across all ozone exposure times, was <10% different from the starting





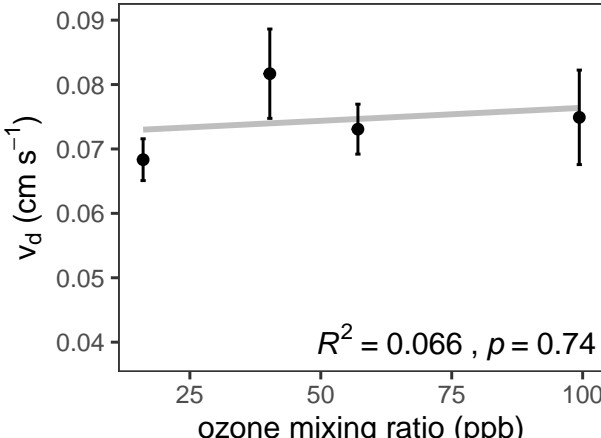

**Figure 4.** Deposition velocity as a function of ozone mixing ratio over iodide in phosphate buffer (10 mM, pH = 8). Mean [I$^-$] = 1.79 μM (1.70, 1.73. 1.93, 1.79 μM, from lowest to highest ozone mixing ratio), T = 298 K.

concentration (Appendix D). For all further analysis, the iodide concentrations reported are the mid-point values, rather than the starting values, to best represent the average conditions of the experiments.

The second order rate constant for ozone with iodide was measured for iodide concentrations between 102 nM and 9.88 μM, and water temperatures between 288 and 303 K (Fig. 5). Results were compiled to an Arrhenius plot (Fig. 6), leading to a calculated pre-exponential factor A of $1.0 \pm 4.6 \times 10^{11}$ M$^{-1}$ s$^{-1}$ and activation energy, $E_a$ of $8.5 \pm 10.9$ kJ mol$^{-1}$. The Pearson correlation coefficient of the Arrhenius plot indicated that the trend was not statistically significant (p = 0.47), therefore the null hypothesis cannot be rejected, i.e. we cannot conclude that the reaction between ozone and iodide is dependent on temperature.






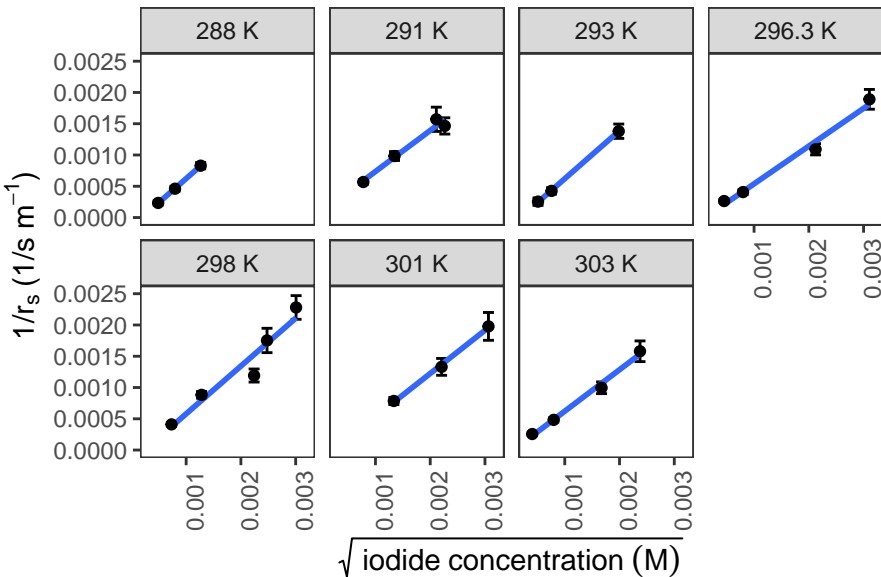

**Figure 5.** Inverse of $r_s$ calculated for various iodide concentrations (102 nM - 9.88 μM), at temperatures between 288 and 303 K. Measurements were performed in triplicate and error bars were propagated from the $r_a$ error, the standard error in linear fit from experimental output, and the errors in measurements of liquid volume and flow tube dimensions.

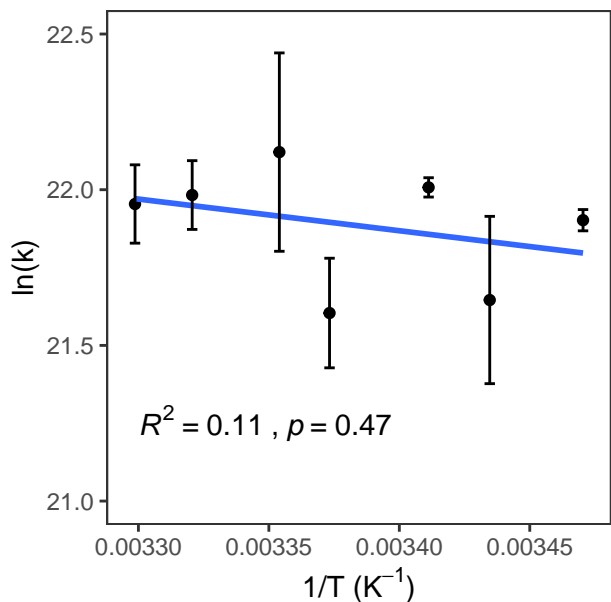

**Figure 6.** Arrhenius plot for reaction between ozone and iodide, the linear correlation has a p = 0.47 and $R^2$ = 0.11. Error bars represent the standard error of the linear fit of $1/r_s$ vs $[I^-]$ for each temperature.





Several studies have measured the second order rate constant at around room temperature, as compiled in Table 1. The rate constants obtained from these studies are plotted as a function of temperature in Fig. 7. Based on experimental conditions, and relevance to marine environments, the studies which are most comparable to ours are those which employ iodide concentrations below $10^{-4}$ M (Moreno et al., 2018); these are Garland et al. (1980); Liu et al. (2001); Shaw and Carpenter (2013). The study by

Liu et al. (2001) was carried out with iodide concentrations approaching the upper limit of where aqueous reactivity dominates, but ozone was applied in solution, removing the possibility of surface reactivity. The results of Garland et al. (1980); Liu et al. (2001); Shaw and Carpenter (2013) report a comparable, but slightly lower $k_{I^- - O_3}$ than this work. A possibility for this could be the lack of replenishment of iodide in their studies. Iodide depletion could occur within the timeframe of their experiments, which would have resulted in a rate of ozone loss lower than anticipated for their expected iodide concentration. Iodide was not

explicitly measured in those studies so any depletion would not be known. Furthermore, the rate constant reported by Garland et al. (1980); Shaw and Carpenter (2013) could be an underestimation due to it not being possible to carry out gas phase diffusion limitation corrections in their experimental setups. For all other reported rate constants, the conditions employed could promote surface reactivity, therefore the measurements are not comparable to the results reported in this work.

**Table 1.** Results and conditions of previous kinetic studies on the reaction between ozone and iodide. "-" denotes condition not reported.

| $k_{I^- - O_3}$ (M$^{-1}$ s$^{-1}$) | Conditions | | | | Method | Reference |
| | T (K) | pH | [I$^-$] | O$_3$ | | |
| --- | --- | --- | --- | --- | --- | --- |
| 2 | 298 | 5.4 | 0.67 - 6.7 $\mu$M | 100 ppb | Stopped flow | (Garland et al., 1980) |
| 4 | 277 | - | 0.5 - 3 M | 7 - 478 ppm* | Droplet train | (Hu et al., 1995) |
| 0.32 - 2.4 | 275 - 293 | - | 0.5 - 3 M | - | Droplet train | (Magi et al., 1997) |
| 1.2 ± 0.1 | 298 | 6.7 | 33.4 - 557 $\mu$M | 27.4 - 40.7 $\mu$M $_{(aq)}$ | Pulse accelerated flow | (Liu et al., 2001) |
| 1.0 ± 0.3 | 293 | - | 7.3 M | 70 - 300 ppb | Aerosol flow tube | (Rouvière et al., 2010) |
| 1.4 ± 0.2 | 293 | 8 | 10 $\mu$M | 70 ppb | Heterogeneous flow reactor | (Shaw and Carpenter, 2013) |
| 3.3 – 3.4 | 288 - 303 | 8.0 | 0.1 - 9.88 $\mu$M | 40 ppb | Heterogeneous flow reactor | This work |

*$5 \times 10^{12}$ – $1 \times 10^{14}$ cm$^{-3}$ ozone reported – converted to ppm (6 – 20 Torr, 277 K)

Of the various reported rate constants for the ozone iodide reaction (Table 1 and Fig. 7), only one other study has explicitly

investigated the temperature dependence and obtained A = $1.4 \times 10^{22}$ M$^{-1}$ s$^{-1}$ and $E_a$ = 73.08 kJ mol$^{-1}$, with an estimated error of 40% (Magi et al., 1997). Our work contradicts the strong positive temperature dependence reported in their work. The difference in conclusion could be due to the differences in conditions. At the concentrations used in our study, bulk reactivity is expected to occur, whereas the conditions employed by Magi et al. (1997) (iodide concentrations up to 3 M) are in a range which could display surface reactivity. The surface reactivity is also dependent on the gaseous ozone concentration, which was

not reported. Further, while interfacial reactivity is not yet fully understood, the pre-exponential factor reported by Magi et al. (1997) is approximately 12 orders of magnitude greater than a diffusion-controlled reaction. In contrast, the pre-exponential factor reported in our work could feasibly be attributed to a diffusion-controlled reaction, within error bounds.



Amalgamating results of single temperature studies by Hu et al. (1995), at 277 K, and the room temperature measurements of Garland et al. (1980); Liu et al. (2001); Rouvière et al. (2010); Shaw and Carpenter (2013), as well as this study, yields a
negligible or slightly negative temperature dependence, within the associated experimental errors, for the bulk phase reaction between ozone and iodide (Fig. 7). Comparing only those experiments which are environmentally applicable, there is no clear trend in temperature dependence. Both conclusions are consistent with the results of our study.

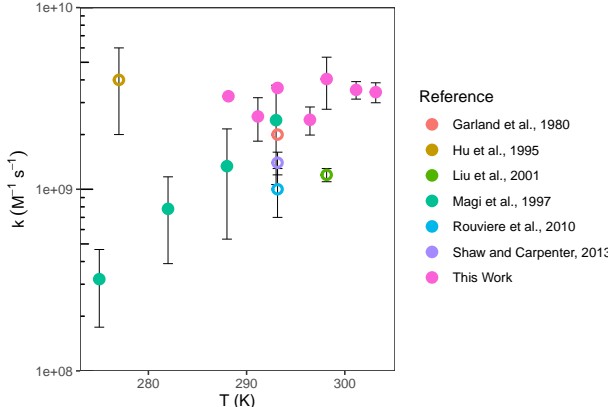

**Figure 7.** Compilation of literature reported second order rate constants between ozone and iodide as a function of temperature. For Hu et al. (1995), errors were not reported - error bars shown are a lower limit estimate based on statements made in the text. Filled circles (●) indicate experiments performed with environmentally relevant conditions. Empty circles (○) indicate experiments which are not environmentally applicable.

### 3.2    Application to Previous Laboratory Measurements of Iodine Emissions

A previous laboratory study of inorganic iodine (HOI and $I_2$) emissions from ozonised iodide solutions ($1 - 5 \, \mu M$ iodide, $222 -$
$3600 \, \mathrm{ppb}$ ozone, temperature $276 - 298 \, \mathrm{K}$) yielded effective activation energies of $17 \pm 50 \, \mathrm{kJ \, mol^{-1}}$ for HOI emission and -7 $\pm 18 \, \mathrm{kJ \, mol^{-1}}$ for $I_2$ emissions (MacDonald et al., 2014). The emissions of HOI and $I_2$ depend on several chemical reactions, each with individual dependencies on temperature, and several physical factors including the solubility and diffusivity of ozone. Thus, the temperature dependence can be negative or positive depending on the combination of these factors. The MacDonald et al. (2014) study was carried out in conditions favouring bulk phase reactions between ozone and iodide, making
their results experimentally comparable to our work. When MacDonald et al. (2014) modelled the emissions of HOI and $I_2$ they demonstrated that their results could only be accurately replicated when assuming $E_a \sim 0 \, \mathrm{kJ \, mol^{-1}}$ for the ozone-iodide reaction, and not when using the Magi et al. (1997) temperature dependence. The model employed by MacDonald et al. (2014), was the sea surface microlayer (SML) model described by Carpenter et al. (2013) except with the inclusion of temperature-dependent processes. The model with temperature dependence did not account for iodine depletion following its fast reaction
with ozone and did not account for iodide replenishment from the waters below (Schneider et al., 2020). To evaluate whether the rate coefficient obtained in our experimental work is consistent with the measured temperature-dependence of fluxes of



gaseous iodine compounds from iodide solution under ozone, we applied our data to an updated SML model (Pound et al., 2023). The updated SML model (details in Appendix E) includes the mixing of iodide between the SML and underlying water and simulates surface iodide depletion, especially at low wind speeds/reduced turbulence.

The SML model was constrained to the range of conditions reported by MacDonald et al. (2014). The ozone mixing ratio and iodide concentration used in our work were 1500 ppb ozone and 2.5 $\mu$M iodide. The effective activation energy was calculated ($E_a = -gradient \times R$) from an Arrhenius type plot of the natural log of the calculated emissions of HOI and $I_2$ (in units of molecules $cm^{-2}\,s^{-1}$) against the inverse of the temperature (in K). It was not possible to accurately calculate an equivalent wind speed for the MacDonald laboratory experiments, therefore two low wind speeds (0.005 and 0.03 $m\,s^{-1}$) were assumed.

The HOI and $I_2$ emissions obtained from the updated SML model are displayed as an effective Arrhenius plot in Fig. 8. The emissions indicate there is a strong dependence of the effective activation energy on the "wind speed" of the experiments, whereby the emissions of HOI and $I_2$ both display a positive effective activation energy at the lowest wind speed, which decreases with the higher wind speed. In the case of $I_2$ this decrease is of a magnitude such that the effective activation energy flips to negative. It is apparent that for a "wind speed" of 0.03 $m\,s^{-1}$, the temperature-dependence of simulated emissions of

both HOI and $I_2$ were within the errors of the experimental measurements of MacDonald et al. (2014). For a "wind speed" of 0.05 $m\,s^{-1}$, the modelled HOI emissions are within the errors of the experimentally determined values of MacDonald et al. (2014) while the effective activation energy of the $I_2$ emissions is slightly overestimated, though captured within our error range.

Knowing that there is negligible temperature dependence on the reaction between iodine and ozone, the relative changes in

each step of the production and emission of HOI and $I_2$ in the Pound et al. (2023) SML model were interrogated to explain the observed wind speed dependencies. Iodide in the surface layer is depleted if the replenishment from the bulk solution occurs at a slower rate than the reaction of iodide with ozone. Iodide depletion was modelled over the range of ozone, iodide and temperature reported by MacDonald et al. (2014), and depletion was predicted to increase with increasing ozone concentration, wind speed and temperature, and with decreasing iodide concentration (Fig. 9). The SML model shows that iodide depletion

increases with increasing ozone concentration and with decreasing iodide concentrations due to chemical consumption of the available iodide. A greater depletion is seen at the higher wind speed because of the increase in ozone deposition as aerodynamic resistance ($r_a$) is reduced with higher airside turbulence ($v_d \approx 0.001014\,cm\,s^{-1}$ at 0.03 $m\,s^{-1}$ and $v_d \approx 0.001002$ $cm\,s^{-1}$ at 0.005 $m\,s^{-1}$). The waterside wind-driven turbulence which could replenish iodide from the bulk is still low at these wind speeds and does not offset the increase in ozone-iodide reactivity, therefore iodide is seen to be more depleted at 0.03

$m\,s^{-1}$ compared to 0.005 $m\,s^{-1}$.

The SML model predicted iodide depletion to increase with temperature due to the slight positive temperature dependence of the ozone-iodide reaction calculated in this work (it was not possible to include error ranges in these calculations). It should be noted that there are very few observations of iodide concentrations at or proximal to the surface layer (Stolle et al., 2020), and it is this iodide which is available for reaction with ozone. Concentrations of iodide at or proximal to the surface may frequently

be different from the reported bulk concentration.

For both wind speeds investigated, ozone deposition velocity was effectively constant over the modelled temperature range (Fig.



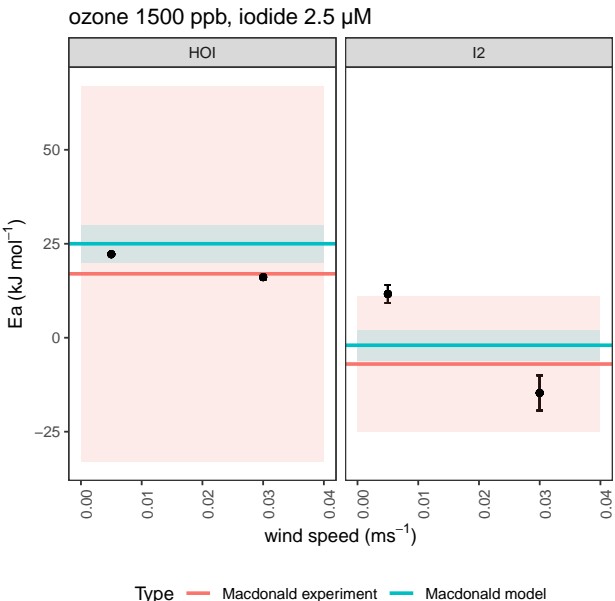

**Figure 8.** Effective activation energies for emissions of HOI and $I_2$ from ozone oxidation (1500 ppb) of iodide solution (2.5 μM) as a function of wind speed. Points show modelled emissions using the SML model (Pound et al., 2023)) while red and blue horizontal lines show experimental and modelled emissions respectively, from MacDonald et al. (2014). Errors are shown by shaded areas, and reflect standard error in the linear fit from modelled output.

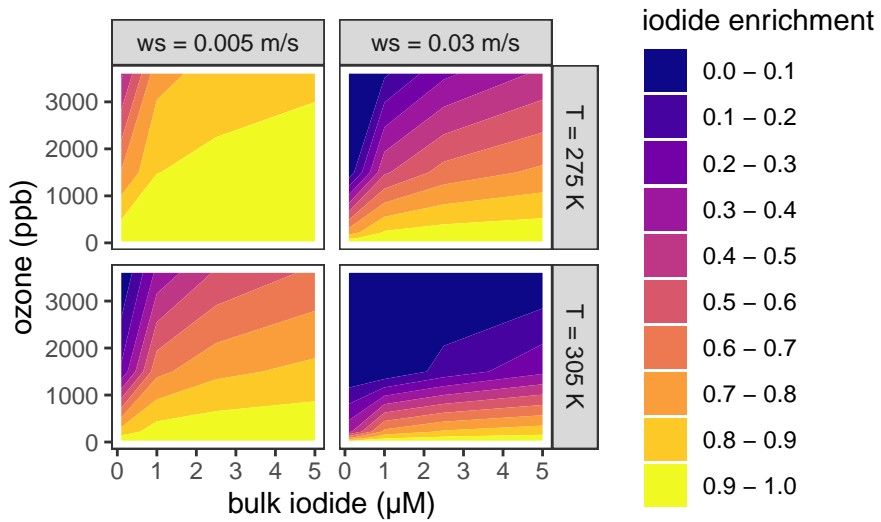

**Figure 9.** Modelled iodide depletion for ozone-oxidised iodide solution, under conditions $1 - 5$ μM [iodide], $222 - 3600$ ppb ozone, T = 276 $- 298$ K. Iodide enrichment = $[iodide]_{SML}/[iodide]_{bulk}$.



10). This is because at low wind speeds, the deposition of ozone is limited by air-side resistance, $r_a$, which has no temperature dependence. Therefore, factors which decrease $r_s$, e.g. changes in ozone solubility, aqueous diffusivity of ozone and reaction rate between ozone and iodide with temperature do not influence deposition significantly. Hence, the availability of ozone in

solution was constant over the prescribed temperature range. Temperature trends in HOI emissions are therefore controlled by drivers in R5, which could include airside transfer velocity $k_a$, and solubility of HOI, expressed as the dimensionless Henry's law coefficient, $H_{HOI}$. The airside transfer velocity shows no variation with temperature, however the solubility of HOI decreases with temperature, leading to greater emissions of HOI at higher temperatures, and hence a positive effective $E_a$. This positive effective $E_a$ is dampened at the higher wind speed by greater iodide depletion in the surface layer at higher

temperature. The overall effect is a less strongly positive effective $E_a$ at $0.03\,\mathrm{m\,s^{-1}}$ than at $0.005\,\mathrm{m\,s^{-1}}$.

For $I_2$ emissions, the formation of $I_2$ is dependent on both HOI and $I^-$ availability (R3, R4). At the lower wind speed, $I^-$ availability is not limiting, and the reduction in HOI availability at higher temperature is offset by the decrease in solubility of $I_2$, resulting in a positive effective $E_a$. At the higher wind speed, the $I^-$ depletion becomes important, and the combined chemical availability of HOI and $I_2$ at higher temperatures limits emissions and causes a negative effective $E_a$ (Fig. 11). The

effects of varying the concentrations of ozone and iodide on activation energy of HOI and $I_2$ emissions were also investigated, and are discussed in Appendix F.

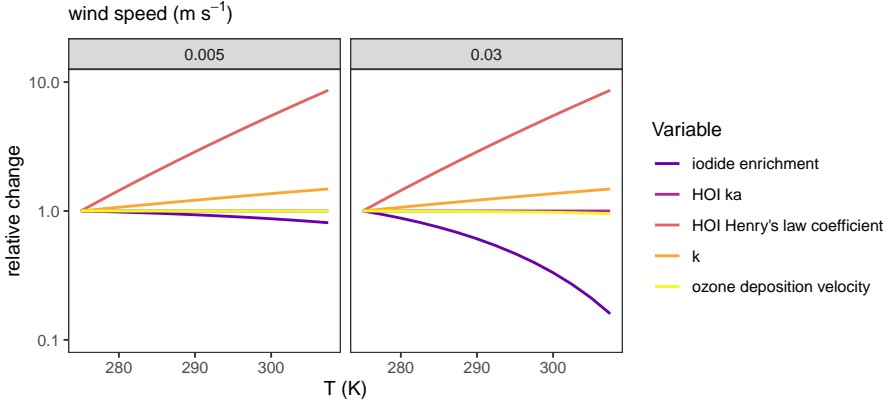

**Figure 10.** Relative change in selected variables with respect to the lowest modelled temperature, for each wind speed. Variables selected with significance for emission of HOI. Model conditions: ozone mixing ratio = 1500 ppb, [iodide] = 2.5 μM.





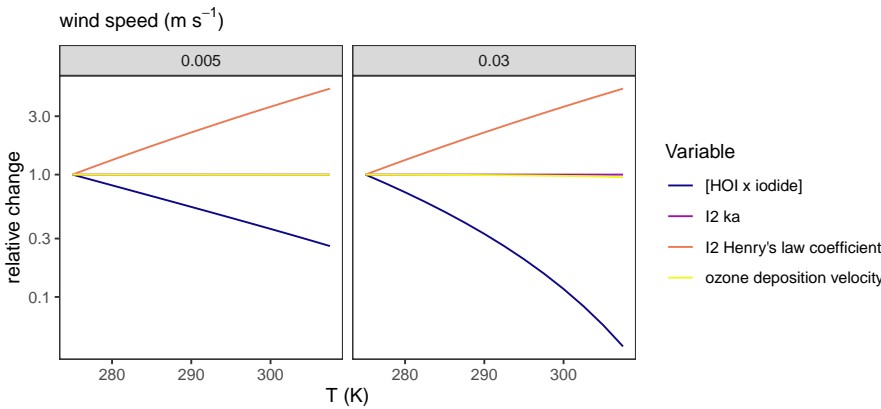

**Figure 11.** Relative change in each variable with respect to the lowest modelled temperature, for each wind speed. Variables selected with significance for emission of $I_2$. Model conditions: ozone mixing ratio = 1500 ppb, [iodide] = 2.5 μM.



## 4 Conclusions

A thorough understanding of the kinetics of the reaction between ozone and iodide in oceanic systems is important for predicting and understanding tropospheric ozone concentrations in remote ocean regions. The second order rate constant of the reaction between ozone and iodide and its temperature dependence were measured in this work using a variable flow methodology, under conditions which emulate the bulk kinetics expected in the surface ocean. A negligible, non-statistically significant temperature dependence was obtained, contradicting a previous study. We therefore conclude that the temperature dependence of this reaction in the ocean has previously been over-estimated.

Though a lack of temperature dependence has previously been implied by comparison between studies, and by back-calculating from emissions, this is the first study, to our knowledge, to show this by direct measurement. The temperature dependence obtained was used to replicate and explore results produced from a previous laboratory study of HOI and $I_2$ emissions from an iodide solution exposed to ozone. This result has implications for oceanic ozone deposition, and emissions of gaseous iodine species both to the troposphere (Carpenter et al., 2013), and stratosphere (Koenig et al., 2020). In related work, we have incorporated these kinetics into a global transport model (GEOS-chem), to improve our understanding of ozone in the troposphere. This work also demonstrates that the laboratory-measured temperature dependence of $I_2$ and HOI emissions, which are a result of complex interactions between physical and chemical parameters, are highly dependent on experimental conditions (including both iodide concentration and ozone mixing ratio), therefore cannot easily be translated into ambient emissions. It is noteworthy that the "wind speeds" applied in our and other authors' experiments are not comparable to those found in the environment. In particular, the very low or negligible water turbulence accessible in laboratory experiments is far lower than those typically found in the ocean. The result of this is that while we simulate increasing iodide depletion in the surface layer with "wind speed" at very low wind speeds, this is not typically expected in the environment where we predict that iodide depletion decreases with increasing wind speed (Pound et al., 2023). The latter is a result of greater wind-induced turbulence increasing the rate of iodide replenishment from the bulk to the surface layer. Therefore, our predicted trends in HOI and $I_2$ laboratory emissions as a function of "wind speed" cannot be directly applied to environmental studies, and this should be considered in experimental design when planning future laboratory work regarding iodine-containing emissions from seawater. Despite this, the good comparability between the modelled results and experimental measurements validates both the kinetic results of this work and the model recently developed by Pound et al. (2023).

## Appendix A: Physical Constants

All physical constants and the values used are outlined in Table A1





**Table A1.** Physical constants used in calculations.

| Constant | Symbol | Value | Unit | Reference |
|---|---|---|---|---|
| Boltzmann constant | $k_B$ | $1.380649 \times 10^{-23}$ | $m^2$ kg $s^{-2}$ $K^{-1}$ | |
| Universal gas constant | R | 8.3145 | J $mol^{-1}$ $K^{-1}$ | |
| Gaseous diffusivity of ozone in air | $D_{g,O3}$ | $0.15 \pm 0.01$ | $cm^2$ $s^{-1}$ | (Langenberg et al., 2020) |
| Temperature | T | 298 | K | |
| Tube length | l | $149.4 \pm 0.1$ | cm | |
| Tube radius | r | $2.02 \pm 0.0203$ | cm | |
| Liquid volume* | liquid volume | $333 \pm 1$ | $cm^3$ | |
| Liquid height * | h | 0.92 | cm | |
| Density of air | $\rho$ | 1.29 | kg $m^{-3}$ | |
| Dynamic viscosity of air | $\eta$ | $1.813 \times 10^{-5}$ | kg $s^{-1}$ $m^{-1}$ | |

* Liquid volume in the flow reactor varied day to day, average values for liquid volume and resulting liquid height are provided here for illustrative purposes, however the measured daily volumes were used in calculations.

## 355 Appendix B: Geometric Equations and Error Calculation

Errors were propagated using the exact formula for the propagation of error (Eq. B1)

$$\sigma_x^2 = (\frac{\delta_x}{\delta_a})^2\sigma_a^2 + (\frac{\delta_x}{\delta_b})^2\sigma_b^2 + (\frac{\delta_x}{\delta_c})^2\sigma_c^2 + ... + (\frac{\delta_x}{\delta_n})^2\sigma_n^2 \tag{B1}$$

### B1 Meta Data

For Eqs. X to X, definitions and values for physical constants can be found in Table A1.

Flow tube volume, $V_{FT}$ (Eq. B2), and associated error (Eq. B3).

$$V_{FT} = \pi r^2 l \tag{B2}$$

$$\sigma_{V_{FT}} = \sqrt{(2\pi rl \cdot \sigma_r)^2 + (\pi r^2 \cdot \sigma_l)^2} \tag{B3}$$

Headspace volume, $V_H$ (Eq. B4), and associated error (Eq. B5).

$$V_H = V_{FT} - liquid\ volume \tag{B4}$$

$$\sigma_{V_H} = \sqrt{\sigma_{V_{FT}}{}^2 + \sigma_{liquid\ volume}{}^2} \tag{B5}$$

Surface area of liquid, SA (Eq. B6), and associated error (Eq. B7).

$$SA = 2l\sqrt{2rh - h^2} \tag{B6}$$






$$\sigma_{SA} = \sqrt{(2\sqrt{2rh - h^2} \cdot \sigma_l)^2 + (\frac{2lh}{\sqrt{2rh - h^2}} \cdot \sigma_r)^2 + (\frac{2l(2r - 2h)}{2\sqrt{2rh - h^2}} \cdot \sigma_h)^2} \qquad (B7)$$

**B2    Data Analysis**

**Appendix C:  Gas Diffusion Corrections**

Under laminar flow, molecules in the centre of the flow tube travel faster than molecules at the edges. When trace gas molecules

irreversibly react with the liquid surface, this results in a concentration gradient. The gas flow within the flow tube was determined to be laminar by calculation of the Reynolds number ($R_e$) by Eq. C1, where $r_h$ is the hydraulic radius of the flow tube, $\rho$ is the density of air, $\eta$ is the dynamic viscosity of air (values in Table A1) and $\nu$ is the velocity of the gas flow in m s$^{-1}$. $r_h$, calculated by Eq. C2 from the area and perimeter of the headspace, must be used as the headspace is not perfectly cylindrical due to the presence of the liquid phase segment. The headspace area is calculated by Eq. C3, where r is the radius of the flow

tube, and h is the liquid height. The headspace perimeter, Eq. C4, is dependent on the arc length, s, (Eq. C5) and cord length, $\alpha$, (Eq. C6). For these experiments, $R_e$ was between 33 and 377, for flow rates of 0.4 – 4.5 SLPM ($R_e$ < 2000 is considered laminar).

$$R_e = \frac{2r_h\rho\nu}{\eta} \qquad (C1)$$

$$r_h = \frac{headspace\ area}{headspace\ perimeter} \qquad (C2)$$

$$headspace\ area = \pi r^2 - (r^2cos^{-1}(\frac{r - h}{r}) - (r - h)\sqrt{2rh - h^2}) \qquad (C3)$$

$$headspace\ perimeter = s + \alpha \qquad (C4)$$


$$s = 2\pi r - (2rcos^{-1}(\frac{r - h}{r})) \qquad (C5)$$

$$\alpha = 2\sqrt{r^2 - (r - h)^2} \qquad (C6)$$

We accounted for the effects of gas diffusion in our flow reactor using the method of Knopf et al. (2015). These corrections

account for reductions in the mixing ratio of analyte gas in the bulk gas phase, $[O3]_g$, at the reactive surface, leading to a near



surface mixing ratio $[O3]_{gs}$, where $[O3]_{gs} < [O3]_g$. Eq. X describes the correction factor, $C_{g,O3}$, which is the ratio of near-surface ozone concentrations to bulk ozone concentrations, as well as relating the effective (measured) uptake, $\gamma_{eff,O3}$ to the true uptake, $\gamma_{O3}$. $\gamma_{eff,O3}$ and associated error are calculated by Eqs. C8 and C9, from the measured deposition velocity, and $\omega_{O3}$, the thermal velocity of ozone in air, calculated using Eq. C10, where $k_B$ is the Boltzmann constant, T is the temperature

in Kelvin and m is the mass of an ozone molecule (Table A1).

To calculate $\gamma_{O3}$ from $\gamma_{eff,O3}$, Eq. C11 is applied, which uses the Knudsen number, $K_{n_{O3}}$, and effective Sherwood number $N_{Shw}^{eff}$ to account for effects of gas phase diffusion limitations. Systematic errors in $N_{Shw}^{eff}$ and $K_{n_{O3}}$ were determined to be negligible, allowing $3/2 N_{Shw}^{eff} \times K_{n_{O3}}$ to be considered a constant in error analysis, denoted $a$ in Eq. C12, from which error in $\gamma_{O3}$ was calculated.

The Sherwood number describes the ratio of convective mass transfer to the rate of diffusive mass transport. The effective Sherwood number, $N_{Shw}^{eff}$, can be approximated by Eq. C13, where A = 0.0978, B = 0.0154 and l* is the dimensionless axial distance calculated by Eq. C14, normalising the axial distance, l (= tube length) using the ratio of $D_{g,O3}$ to volumetric flow rate, Q. As $N_{Shw}^{eff}$ is dependent on Q, which varies throughout the experiment, it was investigated whether the varying flow rates used in these experiments would require separate treatment in gas phase diffusion corrections. For experimental flow rates

of between 1600 – 4500 SCCM, $N_{Shw}^{eff}$ was calculated as 3.73 – 3.86. Due to the small variation in $N_{Shw}^{eff}$, it was decided to apply the mean value ($N_{Shw}^{eff}$ = 3.78) in all instances to simplify calculations. Error in $N_{Shw}^{eff}$ is reported as half of the range of the 5 possible values corresponding to each experimental flow rate, $\sigma_{N_{Shw}^{eff}}$ = 0.06.

The Knudsen number ($K_{n_{O3}}$), Eq. C15, is a unitless dimension describing the ratio of the mean free path of a molecule compared to the hydraulic diameter of the flow tube (Eq. C16). The mean free path of an ozone molecule $\lambda_{O_3}$ was calculated

using Eq. C17, where $D_{g,O3}$ is the gas phase diffusion coefficient. $K_{n_{O3}}$ for this system was calculated as $\approx 6.5 \times 10^{-6} \pm 1.7 \times 10^{-7}$ (reported error is standard deviation across all measurement days). $K_{n_{O3}} \ll 1$ can indicate the flow could be considered as a continuum, except in situations where the uptake coefficient is low and is of a similar magnitude as $K_{n_{O3}}$. For all experiments reported herein, $\gamma_{O3}$ is of the same order of magnitude as $K_{n_{O3}}$, therefore gas phase diffusion corrections must be carried out, using Eq. C11.


$$C_{g,O3} = \frac{[O_3]_{gs}}{[O_3]_g} = \frac{\gamma_{eff,O3}}{\gamma_{O3}} \qquad (C7)$$

$$\gamma_{eff,O3} = \frac{4v_d}{\omega_{O3}} \qquad (C8)$$

$$\sigma_{\gamma_{eff,O_3}} = \frac{4}{\omega_{O3}} \sqrt{\sigma_{vd}{}^2} \qquad (C9)$$


$$\omega_{O3} = \sqrt{\frac{3k_B T}{m}} \qquad (C10)$$



$$\gamma_{O3} = \frac{\gamma_{eff,O3}}{1 - \gamma_{eff,O3}\left(\frac{3}{2 \cdot N_{Shw}^{eff} \cdot K_{n_{O3}}}\right)} \tag{C11}$$

$$\sigma_{\gamma_{O3}} = \sqrt{\left(\frac{\gamma_{eff,O3}^{-2}}{(\gamma_{eff,O3}^{-1} - a)^2} \cdot \sigma_{\gamma_{eff,O_3}}\right)^2} \tag{C12}$$

$$N_{Shw}^{eff} = 3.6568 + \frac{A}{l* + B} \tag{C13}$$

$$l* = \frac{\pi D_{g,O3} l}{2Q} \tag{C14}$$

$$K_{n_{O3}} = \frac{2\lambda_{O_3}}{hydraulic\ diameter} \tag{C15}$$

$$hydraulic\ diameter = 4 \times r_h \tag{C16}$$

$$\lambda_{O_3} = \frac{3D_{g,O3}}{\omega_{O3}} \tag{C17}$$

## Appendix D: Iodide concentrations measured by IC

Iodide concentrations before and after passing through the flow tube are outlined in Table D1, as well as calculated $r_s$ at each iodide concentration and temperature.





**Table D1.** Iodide concentrations before and after passing through the flow reactor, and associated $r_s$ measurements. "-" denotes measurement not available.

| T (K) | $[I^-]$ before (M) | $[I^-]$ after (M) | $[I^-]$ % change | $r_s$ (s cm$^{-1}$) | $r_s$ error (s cm$^{-1}$) |
|---|---|---|---|---|---|
| 288.15 | $1.73\times10^{-6}$ | $1.60\times10^{-6}$ | -7.4 | 12.05 | 0.45 |
| 288.15 | $6.14\times10^{-7}$ | $6.35\times10^{-7}$ | 3.4 | 21.69 | 1.36 |
| 288.15 | $2.37\times10^{-7}$ | $2.39\times10^{-7}$ | 0.7 | 42.81 | 4.99 |
| 291.15 | $5.29\times10^{-6}$ | $5.14\times10^{-6}$ | -3.0 | 6.82 | 0.31 |
| 291.15 | $4.48\times10^{-6}$ | $4.46\times10^{-6}$ | -0.5 | 6.36 | 0.39 |
| 291.15 | $1.92\times10^{-6}$ | $1.82\times10^{-6}$ | -5.1 | 10.15 | 0.44 |
| 291.15 | $6.64\times10^{-7}$ | $6.06\times10^{-7}$ | -8.7 | 17.58 | 0.76 |
| 293.15 | $5.81\times10^{-7}$ | $5.76\times10^{-7}$ | -0.8 | 23.47 | 2.19 |
| 293.15 | $2.82\times10^{-7}$ | $2.62\times10^{-7}$ | -7.2 | 39.30 | 7.93 |
| 293.15 | $4.13\times10^{-6}$ | $3.94\times10^{-6}$ | -4.5 | 7.24 | 0.31 |
| 296.45 | $9.62\times10^{-6}$ | $9.65\times10^{-6}$ | 0.3 | 5.29 | 0.19 |
| 296.45 | $4.55\times10^{-6}$ | $4.51\times10^{-6}$ | -0.9 | 9.16 | 0.43 |
| 296.45 | $6.34\times10^{-7}$ | $6.38\times10^{-7}$ | 0.5 | 24.57 | 1.89 |
| 296.45 | $2.27\times10^{-7}$ | $2.05\times10^{-7}$ | -9.5 | 37.90 | 2.36 |
| 298.15 | - | $5.02\times10^{-6}$ | | 8.39 | 0.41 |
| 298.15 | $6.53\times10^{-6}$ | $6.13\times10^{-6}$ | -6.0 | 5.70 | 0.29 |
| 298.15 | $1.73\times10^{-6}$ | $1.64\times10^{-6}$ | -5.0 | 11.32 | 0.42 |
| 298.15 | $5.46\times10^{-7}$ | $5.37\times10^{-7}$ | -1.6 | 24.36 | 1.13 |
| 298.15 | $8.98\times10^{-6}$ | $9.05\times10^{-6}$ | 0.7 | 4.38 | 0.14 |
| 301.15 | $9.88\times10^{-6}$ | $9.39\times10^{-6}$ | -5.0 | 5.05 | 0.25 |
| 301.15 | $5.19\times10^{-6}$ | $4.88\times10^{-6}$ | -6.0 | 7.51 | 0.41 |
| 301.15 | $1.90\times10^{-6}$ | $1.78\times10^{-6}$ | -5.9 | 12.73 | 0.60 |
| 301.15 | $6.74\times10^{-7}$ | - | | 21.38 | 1.17 |
| 303.15 | $5.75\times10^{-6}$ | $5.64\times10^{-6}$ | -2.0 | 6.32 | 0.33 |
| 303.15 | $6.33\times10^{-7}$ | $6.38\times10^{-7}$ | 0.7 | 20.70 | 1.07 |
| 303.15 | $1.82\times10^{-7}$ | $1.65\times10^{-7}$ | -9.3 | 38.98 | 2.53 |
| 303.15 | $2.97\times10^{-6}$ | $2.80\times10^{-6}$ | -5.9 | 10.03 | 0.56 |



## Appendix E:  Model Description

This model was developed for prediction of ozone deposition to the SML and calculation of subsequent emission of halogenated species. It was designed for environmental conditions, however has been adapted and applied to lab experiments over iodide solutions for the purpose of this work. This model was developed in Python using Cantera as the chemistry solver (Goodwin et al., 2022). The model presented here also uses functions from SciPy (Virtanen et al., 2020), Pandas (pandas development team, 2020), and NumPy (Harris et al., 2020). A summary of the model is included below, but for a complete description and

characterisation of the model, see Pound et al. (2023).

In the model, ozone dry deposition velocity ($v_d$) is calculated using the resistance-in-series scheme (Wesely and Hicks, 1977), which calculates the flux of ozone into the ocean surface microlayer. Airside resistances that represent turbulent transport to the surface and transport through the quasilaminar sub-layer, which is the air directly in contact with the surface microlayer are calculated from wind speed, friction velocity, and the Schmidt number of ozone in air (Chang et al., 2004). The surface

resistance ($r_s$) is calculated using the two-layer method of (Luhar et al., 2018), from the dimensionless solubility, the chemical reactivity, the diffusivity in water, the water-side friction velocity, the thickness of the reaction-diffusion layer of the sea-surface microlayer, and modified Bessel functions of the second kind with order zero and one respectively. $v_d$ is coupled to the SML chemistry via $I^-$ concentration and is recalculated as the model advances towards equilibrium.

This model focuses on the aqueous inorganic halogen chemistry in the SML, applying an extended set of inorganic iodine

chemistry compared to that described by Carpenter et al. (2013). The rate constant used for the reaction between ozone and iodide is that which was calculated in this work. The net flux of $I_2$ and HOI into the atmosphere is calculated from the concentration in the liquid surface and the concentration in the atmosphere, along with the dimensionless Henry's law coefficient for each species, the friction velocity, drag coefficient, Schmidt number and von Karman constant.

This model accounts for mixing from the surface into underlying solution and follows the form of approach used by Cen-Lin

and Tzung-May (2013). The first of these (molecular transfer) is calculated from the waterside transfer velocity and the bulk and surface concentrations of the species in question. In this model there is the facility to account for the effects of surfactants, however we expect no surfactant effect in this work, so this was turned off. The second process, mixing from surface renewal, is a significantly slower process than the mixing described above and is typically on the order of several minutes, but has been included for completeness.

Conditions were chosen to mimic the experiments described by MacDonald et al. (2014). The model is "buffered" to pH 8 by manually resetting $H^+$ and $OH^-$ at each time-step to maintain a constant pH. For ozone mixing ratio and iodide concentrations, a range of values was reported. For the analysis in the main paper intermediate values were used (1500 ppb ozone, 2.5 $\mu$M iodide), however the top and bottom limits of these ranges were also investigated, and are discussed below.



## Appendix F: Modelled $I_2$ and HOI Emissions

### F1 Effect of ozone mixing ratio on Ea

At lower ozone mixing ratios, the iodide is not depleted in the surface (Fig. 9), therefore positive activation energies in HOI emissions (Fig. F1) are driven by reduced solubility of HOI at higher T (Fig. F2). As the $O_3$ mixing ratio increases to 1500 ppb the iodide is depleted at higher temperatures, limiting formation of HOI reducing the Ea. For $I_2$ emissions, the limited HOI production at higher T and $O_3$ mixing ratio is compounded by depletion of iodide, which limits formation of $I_2$, and negative activation energies are modelled.

### F2 Effect of iodide concentration on Ea

Figure F3 shows activation energies of HOI and $I_2$ emissions at different iodide concentrations. At the lowest iodide concentration iodide is depleted in the surface layer to essentially zero at higher temperatures (Fig. 9), causing ozone deposition to the solution to drop. The limited ozone and iodide reduced HOI in the surface layer at higher temperatures, which is not entirely offset by its decrease in solubility, therefore the activation energy for HOI at the lowest iodide concentration is slightly negative (Fig. F4). As iodide concentration increases, iodide depletion decreases, and ozone deposition velocity is almost constant over the temperature range. Iodide is also less depleted in the surface, therefore there is more available HOI, and the positive activation energies seen are due to the reduced solubility of HOI at higher temperatures.

For $I_2$ emissions, $I_2$ formation at the lowest iodide concentration is limited by availability of ozone, iodide and HOI. The almost complete removal of iodide from the surface layer limits formation of $I_2$ therefore a strong negative activation energy is observed. As the iodide concentration increases, the iodide depletion decreases therefore $I_2$ formation is less limited, though still not offset by the reduced solubility of $I_2$, therefore at the conditions reported, for all studied iodide concentrations the activation energies for $I_2$ emissions are negative (Fig. F5).





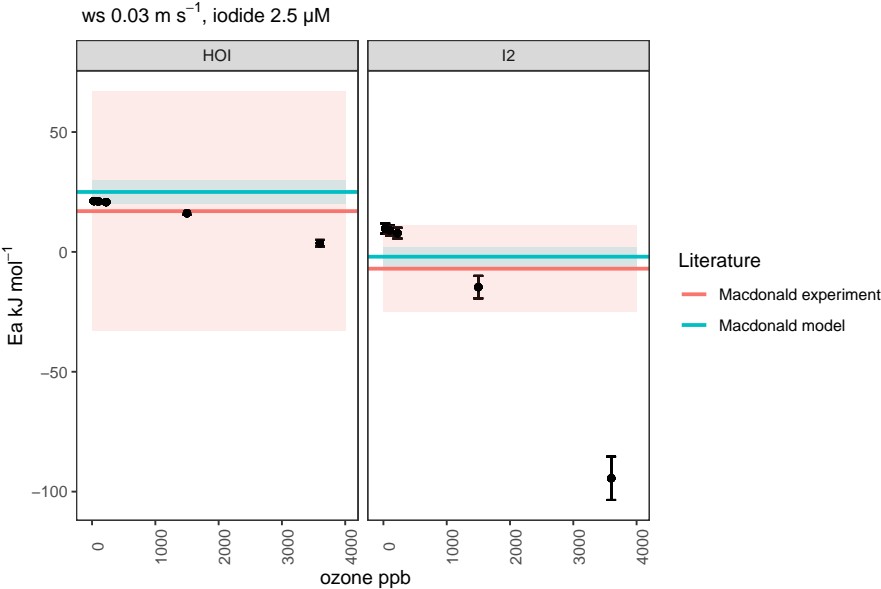

**Figure F1.** Effective activation energies for emissions of HOI and $I_2$ from ozone oxidation of iodide solution (2.5 $\mu$M) as a function of ozone mixing ratio at 0.03 m s$^{-1}$ wind speed. Points show modelled emissions the SML model (Pound et al., 2023), while red and blue horizontal lines show experimental and previously modelled emissions respectively (MacDonald et al., 2014). Errors are shown by shaded areas, and reflect standard error in the linear fit from modelled output.

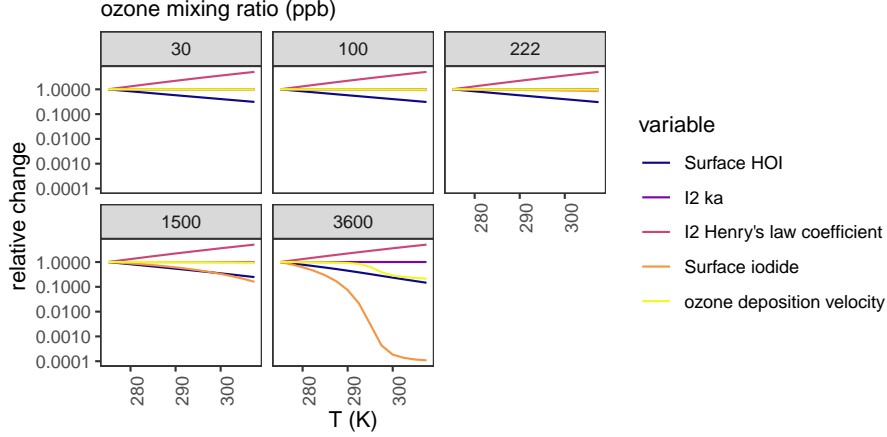

**Figure F2.** Relative change in selected variables with respect to the lowest modelled temperature, for each ozone mixing ratio. Model conditions: wind speed = 0.03 m s$^{-1}$, [iodide] = 2.5 $\mu$M.



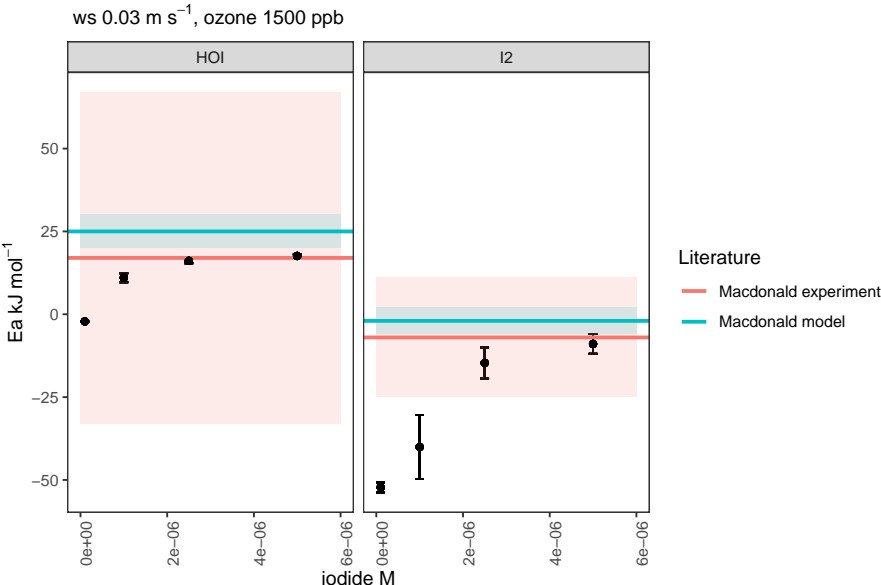

**Figure F3.** Effective activation energies for emissions of HOI and $I_2$ from ozone oxidation (1500 ppb, wind speed 0.03 m s$^{-1}$) of iodide solution, as a function of iodide concentration. Points show modelled emissions using the SML model (Pound et al., 2023), while red and blue horizontal lines show experimental and previously modelled emissions respectively, from MacDonald et al. (2014). Errors are shown by shaded areas, and reflect standard error in the linear fit from modelled output.

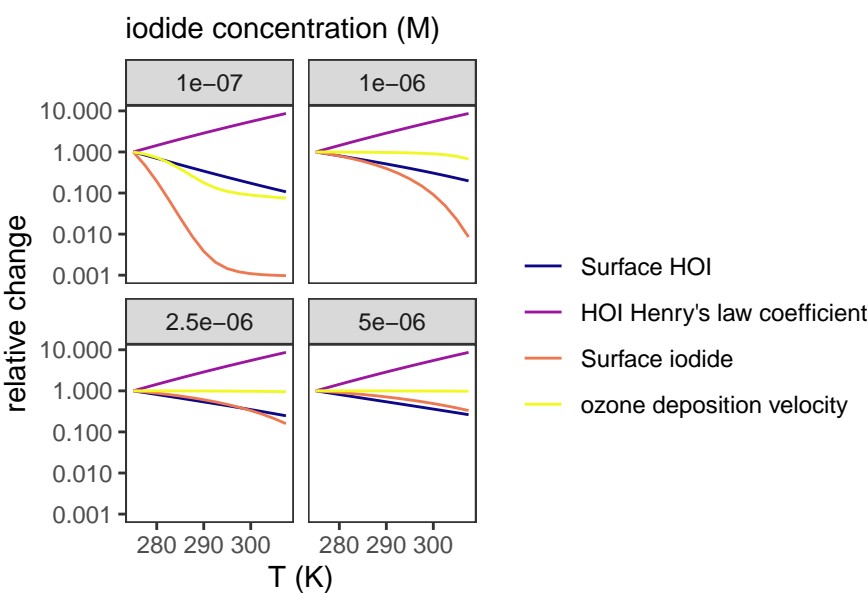

**Figure F4.** Relative change in each variable with respect to the lowest modelled temperature, for each iodide concentration. Variables selected with significance for emission of HOI. Model conditions: ozone mixing ratio = 1500 ppb, wind speed = 0.03 m s$^{-1}$.

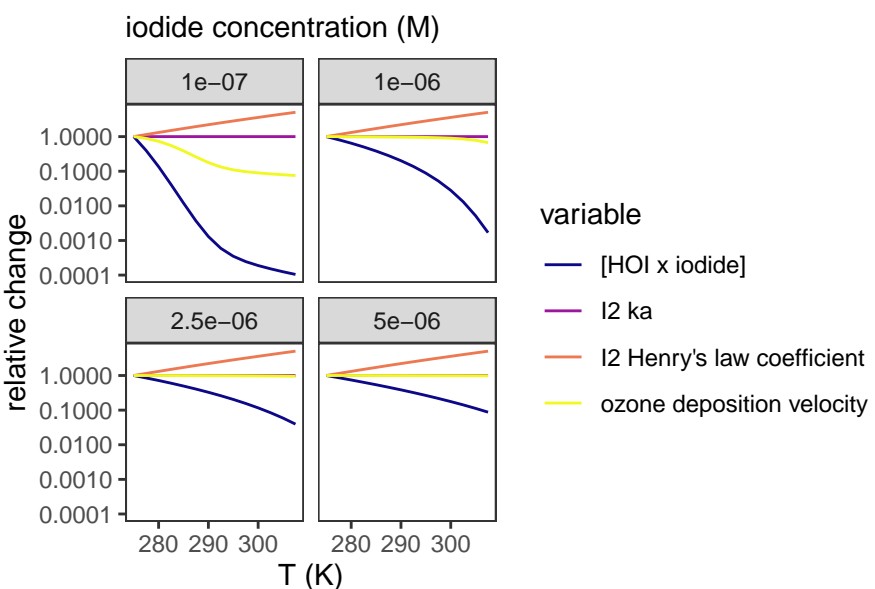

**Figure F5.** Relative change in each variable with respect to the lowest modelled temperature, for each iodide concentration. Variables selected with significance for emission of $I_2$. Model conditions: ozone mixing ratio = 1500 ppb, wind speed = 0.03 m s$^{-1}$.



*Author contributions.* LVB carried out laboratory work, analysed model output and prepared the manuscript. RJP carried out modelling work. LSE assisted with laboratory work. MRJ carried out iodide quantification. SJA and LVB developed the laboratory method. LJC oversaw laboratory and modelling work, and manuscript preparation.

*Competing interests.* The authors are not aware of any competing interests.

*Acknowledgements.* We are grateful for funding from the European Research Council (ERC) under the European Union's Horizon 2020 programme (grant agreement no. 833290).



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
