# Peer review of "Negligible Temperature Dependence of the Ozone-Iodide Reaction and Implications for Oceanic Emissions of Iodine"

_EGUsphere, 2023_

## Author Response (AR1)

**Authors' Comment**

We would like to thank all three anonymous reviewers for their encouraging and constructive feedback. We appreciate their contribution to improving the quality of this paper.

All three reviewer reports are compiled in the attached PDF in black font, and our response to each comment is displayed below, indented and in blue font. Where the manuscript has been altered, line numbers and page numbers are given, and refer to the updated manuscript (without tracked changes).

Upon review we have noticed a small error in the initial submission; the aerodynamic resistance, $r_a$, was measured and subtracted from the iodide measurements, and both the $r_a$ and iodide measurements were corrected for resistance to gas diffusion. This meant two corrections were applied for diffusion resistance. The correct procedure is instead to either estimate $r_a$ using the Knopf (2015) method, or to measure $r_a$ and subtract it from the iodide measurements. We have opted to perform the latter; therefore discussion and explanation of the Knopf (2015) method has been removed, including Appendix section C.  The impact on subsequent calculations is small, resulting in $E_a$ = 7.0 ± 10.5 kJ mol$^{-1}$ and $A$ = 5.4 ± 23.0 × 10$^{10}$ M$^{-1}$ s$^{-1}$ (updated from $E_a$ = 8.5 ± 10.9 kJ mol$^{-1}$ and $A$ = 1.0 ± 4.6 × 10$^{11}$ M$^{-1}$ s$^{-1}$). The conclusions of the paper regarding kinetics remain unchanged.

An updated version of the SML model has also been released during review of this current manuscript, which fixed a bug involving calculation of the SML depth, where the model was not selecting the correct $O_3$+I$^-$ rate. Both the Magi et al., (1997) and Brown et al., (2023) rate coefficients exist as options in the model, however the Magi et al. (1997) rate was always being applied in calculation of SML depth.  As the Magi rate is slower than the Brown rate at relevant temperatures, this resulted in a deeper SML, impacting mixing and species concentrations, specifically, underestimating the iodide depletion in the SML. We have chosen to re-run our modelling section with the latest release, and this has had an impact on the modelled results (the impact of the updated kinetics on the modelling output was small, but is included in the re-runs), therefore discussion section 3.2 has undergone more significant re-writing, with changes to the explanations of driving forces behind temperature impacts on emissions. The impact of iodide and ozone concentrations was previously included as Appendix F, however all discussion of these impacts has been moved to the main text (section 3.2). This includes the addition of Figures 12 and 13, which are updated versions of previous figures F3 and F5. Figures F1, F2 and F4 are no longer required for discussion so have been omitted from the updated manuscript.

The updates to the model are documented here https://github.com/r-pound/COAGEM

**Anonymous Reviewer 1**

This paper addresses the temperature dependence of the kinetics of the reaction between ozone and iodide dissolved in water, i.e., the reaction that dominates the dry deposition velocity of ozone to the ocean surface in the absence of high concentrations of dissolved reactive organics. This reaction also leads to the release of iodine to the atmosphere. The overall reaction rate is sufficiently slow, because of low dissolved iodide concentrations, that the aerodynamic resistance for ozone uptake is relatively minor, and so the rate constant of the reaction itself is particularly relevant for the atmosphere.

There is one reported temperature dependence for the rate constant of this reaction, by Magi et al., with a very high activation energy and pre-exponential factor. The present study finds minimal (potentially nil) temperature dependence to the rate constant, which has direct impact on the modeling of ozone dry deposition and iodine release to the atmosphere. The Magi et al. study was performed with high iodide concentrations (up to molar levels), roughly a million or more times higher than those in the ocean. Thus, the Magi et al. kinetics are almost certainly dominated by interfacial processes whereas the oceanic reaction occurs in the bulk, away from the interface, as delineated in a couple of papers by Moreno and co-workers. So, the kinetic parameters of Magi et al. are not relevant (nor are they likely correct, i.e., a rate constant that is so close to the diffusion limit cannot have such a large activation energy). Rather, the present study was conducted with atmospherically relevant concentrations of ozone and iodide. The finding of a minimal temperature dependence is not surprising given the size of the rate constant (i.e., approaching the diffusion limit) but nevertheless important to confirm.

I recommend publication after the following comments are addressed:

1. Reaction 1 likely has a reversible (i.e., double arrowed) reaction of ozone and iodide to form IOOO-; after formation, this complex can either decompose back to reactants or go on to form IO- and O2. In this context, how does the minor temperature dependence in R1 arise? Is the rate determining step the formation of IOOO-, with a low barrier in the entrance channel to the reaction? Or, if the complex lives for a reasonably long time, how would the balance between the forward and backward reactions of IOOO- affect the temperature dependence for the overall reaction? The potential atmospheric significance is if something else in solution may affect the fate of IOOO-, such as dissolved organics.

We have added a comment in the text to explain that the backwards path of complex decomposition of IOOO⁻ is thermodynamically unfavourable, and added a reversible arrow to R1 for completeness. (**page 2, lines 52 – 55**)

We have added some discussion to section 3.1 (**page 14, lines 276 to 283**) of the theoretical activation energies of the formation of IOOO⁻ , which has been suggested as the rate limiting step of the ozone+iodide reaction and calculated in separate studies as either weak or strongly positive.  Our study supports an even weaker barrier.

In terms of atmospheric significance, subsequent reactions of IOOO⁻ do not affect ozone uptake, but impact the speciation of total inorganic iodine emissions. This is intended to be explored in future development of the SML model, therefore we have chosen not to discuss this in the current manuscript.

2.  The care taken to avoid and assess iodide depletion is commendable. Indeed, the overall experimental setup is nicely configured and explained.

    The authors thank the reviewer for your comments.

3.  Although the primary media was ozonized to remove potential contaminants, was there any evidence for the tubing or pump that feed the iodide solution to the flow tube to be a source of reactive contaminants?

    We did not have any reason to suspect reactive contaminants from the tubing to the solution. The tubing used was Marprene, which is resistant to chemical degradation by acid and ozone therefore we expect no breakdown of the material. It was pumped using a peristaltic pump therefore there was no contact with the pumping mechanism to be a source of contaminants. The blank solution was continuously passed through the reservoir, tubing and flow reactor during pre-ozonisation and during blank measurement, therefore reactive contaminants, if any, would be pre-ozonised and accounted for in the blank measurement. Cleaning media (HCl and milli-Q) were also passed through the tubing into the flow reactor, therefore the tubing was rinsed to the same extent as the flow reactor.

4.  Would the kinetics have been different with seawater concentrations of salts, in particular chloride? I know that ozone does not rapidly react with chloride but could the ionic strength of seawater affect the kinetics and its temperature dependence?

Moreno et al. (*A Kinetic Model for Ozone Uptake by Solutions and Aqueous Particles Containing I- and Br-, Including Seawater and Sea-Salt Aerosol*, Phys. Chem. Chem. Phys, 2019) hypothesised that the presence of chloride could catalyse the surface reaction of ozone and iodide. However, as the reaction occurs in the bulk phase under the ambient conditions used in our study, we do not believe this is influencing our results. Note also that Schneider et al. (*Iodine emission from the reactive uptake of ozone to simulated seawater*, Environmental Science: Processes and Impacts, 2022) observed experimentally that the addition of NaCl did not impact ozone loss to KI solution.

Chloride does impact speciation of iodine emissions, therefore its impact is included in the SML model, however in this work the experiment we attempt to replicate did not include any chloride.

5. Minor superscript and subscripts issues, e.g., lines 147, 179 and a number of other places where the 3 is O3 is not a subscript.

Updated throughout text.

6. The authors subtract the loss observed in blank experiments (without iodide present) from the results for iodide experiments. This assumes that the loss observed in the blank also occurs when iodide is present. I don't know if this is necessarily true, i.e., without iodide present, ozone will diffuse deep into the solution until it finds something to react with. When iodide is present, ozone is constrained to the reacto-diffusive depth of the iodide solution (a few microns) and is less likely to react with those contaminants. So, it is possible that the blank should not be subtracted from the observed kinetics with iodide present. How much would the results of the paper be changed if the blank is not subtracted?

We follow a similar approach to that used by Schneider et al., 2020 (Reactive Uptake of Ozone to Simulated Seawater: Evidence for Iodide Depletion), whereby the blank measurement is simply subtracted from the iodide measurement. In Schneider's work, the experimental data with background correction was well aligned with model predictions of their system (when gas-phase diffusion corrections were included in the model), while the measurements without background correction overestimated the uptake coefficient compared to the model (both with and without diffusion corrections).

We believe that, even if there is a small influence of extra reactivity in the blank, the blank has to be included to account for any contamination from the phosphate buffer, and ozone losses to the glass surface, fittings and tubing. Note that if contaminants are

organic, we would expect any reaction with ozone to occur on the liquid surface, therefore these would have no influence on reacto-diffusive length.

We cannot quantify this impact, however we have added a statement to the paper with a caveat that if loss to blank is not making a large contribution to the loss of ozone when iodide is present then $A$ and $E_a$ are underestimated (**page 11, lines 248 – 251**).

If we do not include the blank measurement, the pre-exponential factor $A$ increases to $2 \pm 16 \times 10^{14}$ $M^{-1}$ $s^{-1}$ (with $E_a$ increased to $24.6 \pm 20.3$ kJ mol$^{-1}$); a rate beyond the limit of a diffusion controlled reaction, an unphysical value.

7. The paper claims a connection to the lower stratosphere, where iodide oxidation may be occurring. I can see the potential connection but the temperature range investigated in this work is small when thinking about stratospheric conditions. Perhaps add a minor caveat?

We have added a comment to the conclusion to clarify the differences in temperature ranges, which reads "Despite being outside of the temperature range studied here, this work has potential further implications for halogen emissions to the stratosphere." (**page 22, lines 399 – 400)**

8. The open and closed symbols in Figure 7 are a bit confusing to me. For example, the data from Magi et al. have closed circles but the authors claim that that study is not environmentally relevant (and closed circles are meant to indicate environmental relevance) whereas the data from Garland et al. are open circles but their work is environmentally relevant …

The open/closed circles on this graph were not correct in the original manuscript, and did not correspond to the statements in the text. Figure 7 has been corrected in the revised manuscript.

9. Which rate constant is the "k" in the legend of Figure 10? First or second order? Also, the "ka" for HOI should be "Ka".

The figure legend has been updated to specify this is $k_{O3\text{-}I\text{-}}$. $k_a$ (small k) refers to the airside transfer velocity, defined in text, similar for $k_w$. (**page 15**, **line 323 and page 17, line 344**).

10. Concerning the broader question of the temperature dependence of iodine release driven by iodide ozonolysis, I am guessing that one of the strongest uncertainties will be the uncertainties in the temperature dependencies of the Henry's Law constants for HOI (and I2). Is that true? I didn't see this point made in the paper, but I may have missed it. Figure 10 illustrates the strong dependence on the HOI Henry's Law constant but what are the uncertainties in that temperature dependence?

> The temperature dependence in Henrys law constants included in these calculations are estimates obtained using equations from Johnson, 2010 (reference added to text). There are indeed large uncertainties present in calculating gas water transfer, quoted as up to a factor of 2 in Johnson et al. Due to the lack of accurate quantification for many of these terms and their impact on iodine emissions, these have not been included. We have added a discussion of this uncertainty to the text, (**page 18, lines 381 - 385**).

**Anonymous Reviewer 2**

The manuscript entitled, "Negligible Temperature Dependence of the Ozone-Iodide Reaction. . ." submitted by Brown et al., details a very rigorous study of the aqueous phase reaction rate coefficient of I + O3. The authors have endeavored to determine this rate coefficient under atmospherically relevant concentrations (O3 and Iodide) and temperatures. This fills a gap in previous measurements of this reaction. The authors then used the updated rate coefficients and Arrhenius parameters to constrain a sea surface microlayer model. The importance of this reaction and rate coefficient, as the authors have argued, cannot be understated for understanding marine chemistry.

The experiments are carefully done and described in detail to allow other to replicate their approach. The authors have done a excellent job with experimental checks to minimize iodide depletion in their setup. The authors have done a good job analyzing previous measurements and have done an excellent job explaining where discrepancies exist and why. I recommend publication after the authors have addressed the following:

1. The data points in Figure 3 are hard to read due to the thick lines and the use of yellow. Please revise so a ready can clearly distinguish the data from the fit. Also in the text the residence time is cited as 24-66 seconds whereas in Fig. 3 the first data point is at 20 seconds?

> The times in text were mistaken, they have now been corrected. Note also the in-text equation for reaction time was written the wrong way round, and has also been corrected (**page 7, lines 184-185**). Figure 3

has also updated – the colour scheme and linewidth have been changed to make the figure clearer.

2.  Line 199 For clarity since there has been historical confusion please be explicit about what H =  3.63 means?  i.e. Gas/Aqueous

Clarification added to text, (**page 10, line 210**).

3.  Line 211  There is some discussion about if the mechanism proceeded via Langmuir-Hinshelwood an exponential increase in uptake with decreasing ozone would be observed.  I don't understand this statement and as written it is not clear to me if this statement is in fact correct.  The authors should take a few sentence to explain why this would be expected for a LH mechanism.

We have re-written this section for clarity, reading "Dependence of uptake on the mixing ratio of ozone would be expected if the reaction were proceeding via a surface-mediated Langmuir-Hinshelwood reaction. This is due to surface saturation of ozone at higher mixing ratios limiting potential reactivity on the surface. Therefore, a greater ozone uptake would be expected at lower ozone mixing ratios." (**page 10, lines 222 – 225**).

4.  Table 1.  The rate coefficients in column 1 seem to be missing and 10^9 factor?

Updated in table 1.

5.  Figure 7  The open closed symbols and colors are confusing to denote environmental applicability.  Please revise figure for clarity.

See response to reviewer 1 comment 8.

6.   Line 359  "For Eqs. X to X. . ." is confusing.

This was corrected to read "For Eqs. B2 to B7" (**page 23, line 423**).

7.  The authors should attempt to discuss the negligible (or slightly negative) temperature dependence in light of R1 in particular the formation of intermediate IOOO-.  There have been previous theoretical calculations (see for example Ó. Gálvez, M. Teresa Baeza-Romero, M. Sanz and L. F. Pacios, A theoretical study on the reaction of ozone with aqueous iodide, Physical Chemistry Chemical Physics, 2016, 18, 7651-7660 and others) of this reaction that are useful context for a reader to understand if the temperature dependence observed in the experiment is consistent with theory and the proposed intermediate IOOO-.  In other words, do these new measurements suggest previous theory is or is not correct?

Thank you for drawing our attention to this. A passage has been added to discussion section 3.1 (**page 14, lines 277 to 283**), see also response to reviewer 1 comment 1.

**Anonymous Reviewer 3**

This article titled "Negligible Temperature Dependence of the Ozone-Iodide Reaction and Implications for Oceanic Emissions of Iodine" by Brown et al. presents a thorough measurement of the temperature dependence of the ozone-iodide reaction. This reaction is especially important to understand in the marine troposphere and stratosphere since it drives ozone dry deposition and is the dominant source of gaseous iodine (as HOI and $I_2$) into the atmosphere.

In this study, the authors did an excellent job describing the specific challenges associated with these laboratory measurements. These considerations include the depletion of iodide in the measurement system, making sure to operate at atmospherically relevant concentrations, and the presence of contaminants in their system. Additionally, the authors did a great job in describing the extent and limitations of previous studies measuring similar systems. I recommend publication after the following points are addressed:

1. More details could be provided on the blank measurements and the subsequent iodide experiments. How was the solution ozonized, specifically, was it by bubbling through the solution or from the gas-phase only? Was the iodide spiked directly into the experimental set-up as is suggested on L117 – 118?

   Further details on blank measurement, spiking and iodide measurement have been added to the text (**page 7, lines 185 – 195**).

2. I am a bit confused about the discussion on 'chemical availability' mentioned on L324 which includes: "the combined chemical availability of HOI and $I_2$ at higher temperatures limits the emissions [of $I_2$]... " Does this refer to lower concentrations of HOI and $I_2$ in solution from reduced iodide which results in lower emissions (i.e. R5 and R6)? Or rather, on L321, the authors state "the formation of $I_2$ is dependent on both HOI and $I^-$ availability", so perhaps the statement on L324 contains a typo?

   Yes, this was intended to refer to the low concentrations of $I_2$ and HOI in solution limiting formation and emission of $I_2$. However, in re-writing discussion section 3.2 in light of the above changes, discussion of this parameter has been removed.

3. In Figure 11 and Figure F5, the authors refer to the variable [HOI x iodide] and I'm not sure exactly what this is referring to. Is this the sum of both concentrations? Or is it referring to the reactant availability (from a rate equation) from R3?

> This refers to the reactant availability from R3, the product of both concentrations. As above, this aspect of the discussion has been removed in the updated manuscript.

4. The authors use 'enrichment' to discuss the relative concentration of iodide in the SML relative to the bulk, which is confusing because they are specifically quantifying the depletion in the SML relative to the bulk. Perhaps using the term 'enrichment factor' is clearer since it doesn't immediately suggest that the concentrations are elevated. This also aligns with the terminology used to describe the relative concentration of organics in the SML to the underlying water.

> We agree this is clearer, updated in text (**page 18, line 374**) and figures 9, 10, 11, 13.

5. What is the depth of the SML in the model? Perhaps it is useful to define this, since the SML depth is defined operationally.

> SML depth in the model is defined as the reacto-diffusive depth of ozone, it unique to each combination of conditions. Comment added to text (**page 15, lines 300 – 301**).